# Identification Method of Source Term Parameters of Nuclear Explosion Based on GA and PSO for Lagrange-Gaussian Puff Model

**Yang Zheng [1], Yuyang Wang [2], Longteng Wang [3], Xiaolei Chen [1], Lingzhong Huang [1,4], Wei Liu [1], Xiaoqiang Li [1], Ming Yang [1], Peng Li [1], Shanyi Jiang [1,5], Hao Yin [1], Xinliang Pang [1,* and Yunhui Wu [1,*]

[1] State Key Laboratory of NBC Protection for Civilian, Beijing 102205, China; zheng_yang_rr@163.com (Y.Z.); chenxlei2002@163.com (X.C.); 22140666@bjtu.edu.cn (L.H.); lw_email@126.com (W.L.); lixiaoqiang@sklnbcpc.cn (X.L.); ym_01@126.com (M.Y.); lipeng@sklnbcpc.cn (P.L.); syjiang@njust.edu.cn (S.J.); yinhao@sklnbcpc.cn (H.Y.)

[2] College of Computer and Data Science, Fuzhou University, Fuzhou 350108, China; 221010011@fzu.edu.cn

[3] Peking University-Tsinghua University-National Institute of Biological Sciences Joint Graduate Program, School of Life Sciences, Peking University, Beijing 100871, China; longtengwang@pku.edu.cn

[4] School of Electrical Engineering, Beijing Jiaotong University, Beijing 100044, China

[5] School of Electronic and Optical Engineering, Nanjing University of Science and Technology, Nanjing 210094, China

\* Correspondence: pangxinliang@sina.com (X.P.); www_wu@foxmail.com (Y.W.)

**Abstract:** Many well-established models exist for predicting the dispersion of radioactive particles that will be generated in the surrounding environment after a nuclear weapon explosion. However, without exception, almost all models rely on accurate source term parameters, such as DELFIC, DNAF-1, and so on. Unlike nuclear experiments, accurate source term parameters are often not available once a nuclear weapon is used in a real nuclear strike. To address the problems of unclear source term parameters and meteorological conditions during nuclear weapon explosions and the complexity of the identification process, this article proposes a nuclear weapon source term parameter identification method based on a genetic algorithm (GA) and a particle swarm optimization algorithm (PSO) by combining real-time monitoring data. The results show that both the PSO and the GA are able to identify the source term parameters satisfactorily after optimization, and the prediction accuracy of their main source term parameters is above 98%. When the maximum number of iterations and population size of the PSO and GA were the same, the running time and optimization accuracy of the PSO were better than those of the GA. This study enriches the theory and method of radioactive particle dispersion prediction after a nuclear weapon explosion and is of great significance to the study of environmental radioactive particles.

**Keywords:** genetic algorithm; particle swarm optimization; nuclear explosion; radioactive diffusion

## 1. Introduction

Many people around the world are working tirelessly to prevent the proliferation of nuclear weapons [1,2], but the threat of nuclear weapons still warrants vigilance from all of us [3–5]. In addition to the direct shock wave damage effect and ionizing radiation released by fission, nuclear weapon explosions also produce large amounts of radioactive particles [6–8]. These radioactive particles can contaminate the ground, water, air and the entire public environment, causing significant effects on the surrounding population [9,10]. This process can last for months, years or even longer [7,8,11]. Many nuclear weapons tests throughout history have caused severe radioactive contamination of indigenous populations living near the test sites [12,13], and the surge in the incidence of thyroid cancer and other cancers has further increased the importance of research on radioactive contamination from nuclear explosions [14–19].

After a nuclear weapon explosion, its radioactive particles spread into the surrounding environment with atmospheric motion, and many models have been established to predict the dispersion of its radioactive contaminants [20,21], but the accuracy of various source term parameters, including meteorological information and coordinates of the explosion center, explosion equivalent, etc., is one of the basic and important prerequisites to ensure the accurate input and prediction accuracy of the prediction model. Therefore, it is very necessary to obtain accurate source term parameters through real measurements or optimization algorithms, as well as other means, so as to provide a solid and powerful foundation for the subsequent operations of the prediction model. However, unlike the ideal situation of adequate preparation and accurate source term parameters for nuclear testing, in real situations, especially in the case of a nuclear weapon strike, accurate source term parameters are often difficult to obtain. Therefore, the accuracy of the analysis of the source term parameters of a nuclear weapon explosion determines the accuracy of the prediction of the radioactive dispersion model and the effectiveness of the nuclear emergency response.

In existing studies of nuclear power plant accidents, the uncertainty of source term parameters is one of the main reasons for errors in the evaluation results [22–24]. To obtain more accurate source term parameters for nuclear accidents, techniques to invert the source term parameters of nuclear accidents based on monitoring data combined with artificial intelligence algorithms [25–27], or with swarm intelligence algorithms such as the cuckoo search algorithm [28], GA [29], and PSO [30] have been widely developed. The combination of these optimized algorithms and measured data can provide identification accuracy of source term parameters for accidents such as radioactive leaks in nuclear power plants of more than 95%, which provides a reference for research in the field of nuclear weapon explosions.

In this paper, in order to realize the improvement of the accuracy of the prediction model of the radioactive particle hazard released into the environment after nuclear weapon explosions, the Lagrange-Gaussian puff model is used, and the adaptation function is defined based on group intelligence optimization algorithms, such as the GA and PSO algorithms, combined with monitoring data. After experimenting with the parameter selection of the algorithms, the identification of nuclear weapon explosion source term parameters was optimized, and after testing using real historical data, the PSO algorithm optimized Lagrange-Gaussian puff model was found to be more successful in identifying source term parameters; the prediction accuracy of the main parameters was above 98%. The results have a high degree of confidence and accuracy.

## 2. Materials and Methods

### 2.1. Data Source

All experimental data in this paper are from publicly available historical U.S. nuclear weapons experimental data [31,32]. The data contain sediment data and radiation contour maps for selected atmospheric nuclear explosion experiments in Nevada and Bikini Island from 1945–1962 (with classified material removed), which comply with the requirements of the Atomic Energy Act and are intended to provide a reference for researchers engaged in sediment impact analysis of sedimentation patterns and associated test data. The dataset focuses on recording meteorological parameters such as the time of explosion, explosion equivalent, explosion center coordinates, explosion height, average wind direction and average wind speed at the moment of explosion, and radiation contour maps for one hour after the explosion of the nuclear test. A typical historical atmospheric nuclear explosion experiment was chosen for this study: the nuclear test code-named "KOON" on 6 April 1954 (Figure 1).

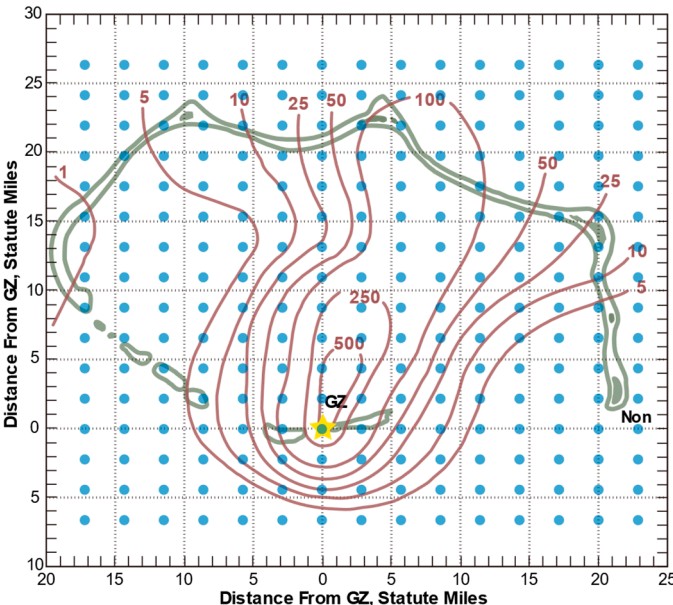

**Figure 1.** One-hour radiation dose contour map in r/h after "KOON" explosion [33].

In this study, the source parameters recorded in the nuclear test "KOON" were used as "true values", as shown in Table 1. Because real source term parameters cannot be accurately obtained in the case of an actual weapon strike, this simulation adds Gaussians to the "True-Value" parameters. Therefore, the simulation experiment uses the "True-Value" parameters based on the addition of Gaussian random error as the initial source data input range, as shown in Table 1. The heart of the of the real explosion used in the experiment had the geographic coordinates of (11°29′48″ N, 165°22′03″ E). Mercator coordinate system data is not easy to calculate, so in this experiment, the explosion heart coordinates were mapped to a plane right-angle coordinate system so that the explosion center coordinates were (1250, 2500) (Table 1).

**Table 1.** The range of values of real source term parameters and primitive source term parameters.

| Source Term Parameters | Parameter True-Value | Gaussian Error | Error Range | Initial Source Term Parameter Range |
|---|---|---|---|---|
| Total Yield (Kt) | 110 | $W = \gamma \cdot W_{true}$ | $\gamma \in [0.1, 10]$ | [11, 1100] |
| Explosive heart x coordinate | 1250 | $x = x_{true} + \Delta x$ | $\Delta x \in [-1250, 3750]$ | [0, 5000] |
| Explosive heart y coordinate | 2500 | $y = y_{true} + \Delta y$ | $\Delta y \in [-2500, 2500]$ | [0, 5000] |
| Average wind speed (m/s) | 6.2 | $v = \lambda \cdot v_{true}$ | $\lambda \in [0.5, 2]$ | [3.1, 12.4] |
| Average wind direction (°) | 191.3 | $\varphi = \varphi_{true} + \Delta \varphi$ | $\varphi \in [-20, 20]$ | [171.3, 211.3] |

The dose on the real 1-hour post-explosion radiation dose contour map is shown in Figure 1 as the "measured value" and will be set up in the region evenly across 250 virtual observation points to obtain the radioactive dose rate data.

## 2.2. Method Process

The methodological flow of this paper is shown in Figure 2. Five important source term parameters that affect the accuracy of the nuclear weapon explosion radioactive particle dispersion model were selected as the source term parameters to be optimized for identification: explosion equivalent, explosion center x, y coordinates, equivalent wind speed and average wind direction.

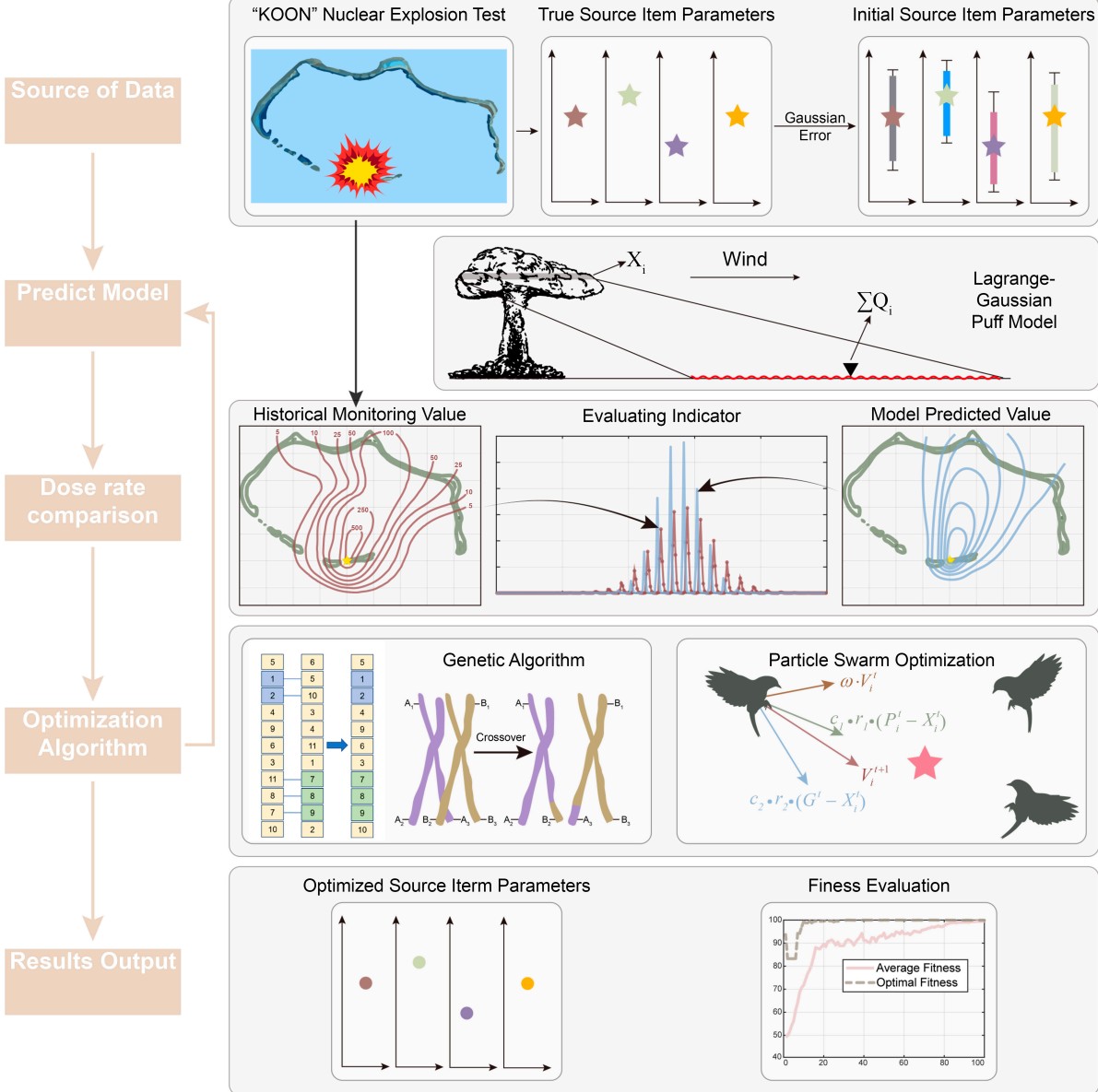

**Figure 2.** Flow chart showing how GA and PSO algorithms are used to optimize the identification of nuclear explosion source term parameters.

The initial random source term parameters with the range of values as in Table 1 were input into the Lagrange-Gaussian puff model to obtain the corresponding model prediction values. Then, their real monitoring values (Figure 1) were compared and fitness was calculated. Then, using the GA and PSO algorithms after the parameter search, after several iterations, an effective and optimal identification of the accurate source term parameters could be achieved (Figure 2).

*2.3. Predictive Modeling of Radioactive Particle Dispersion in Nuclear Explosions*

Many models are available to fit and analyze the physical process of radioactive smoke cloud lifting and particle diffusion and settling with atmospheric motion after a nuclear weapon explosion [20,21]. Among them, the Lagrange-Gaussian puff model is used to simulate the process of diffusion of various pollutants in the atmosphere by tracking the suspended particles that move with the physical motion of the atmosphere [34–36]. This model has remarkable calculation simplicity and adaptation flexibility for a small and medium-scale atmospheric diffusion field [37] and is suitable as a basic model for the

diffusion of radioactive particles after nuclear weapon explosions. The radioactive particles of the nuclear explosion smoke cloud have different sizes and exhibit different settling laws and characteristics, as shown in Figure 3, so the radioactive material released from a nuclear weapon explosion is simulated to be cut into multiple smoke clusters and transported and diffused in the atmosphere by wind. The contribution of each smoke cluster to the concentration of radioactive material at a certain location is expressed as:

$$C_i\left(x_g, y_g, z_g\right) = \frac{A(i)}{(2\pi)^{3/2}\sigma_x(i)\sigma_y(i)\sigma_z(i)} \times \exp\left[-\frac{\left(x_g-x_c(i)\right)^2}{2\sigma_x^2(i)} - \frac{\left(y_g-y_c(i)\right)^2}{2\sigma_y^2(i)}\right] \times$$
$$\left\{\exp\left[-\frac{\left(z_g-z_c(i)\right)^2}{2\sigma_z^2(i)}\right] + \exp\left[-\frac{\left(z_g+z_c(i)\right)^2}{2\sigma_z^2(i)}\right] + \exp\left[-\frac{\left(2z_{inv}-z_c(i)\right)^2}{2\sigma_z^2(i)}\right]\right\}$$

(1)

where $A(i)$ denotes the activity of the radioactive material contained in the ith smoke cluster in Bq; $\left(x_c(i), y_c(i), z_c(i)\right)$ denotes the initial center coordinates of the ith smoke cluster; $\sigma_x(i)$, $\sigma_y(i)$, $\sigma_z(i)$ denotes the diffusion parameters of the ith smoke cluster in horizontal and vertical directions; $z_{inv}$ denotes the height of the atmospheric boundary layer. The contributions of the radioactive material concentrations of all smoke clusters are superimposed to obtain the values of radioactive material concentrations at the specified locations $\left(x_g, y_g, z_g\right)$.

$$C\left(x_g, y_g, z_g\right) = \sum_i C_i\left(x_g, y_g, z_g\right)$$

(2)

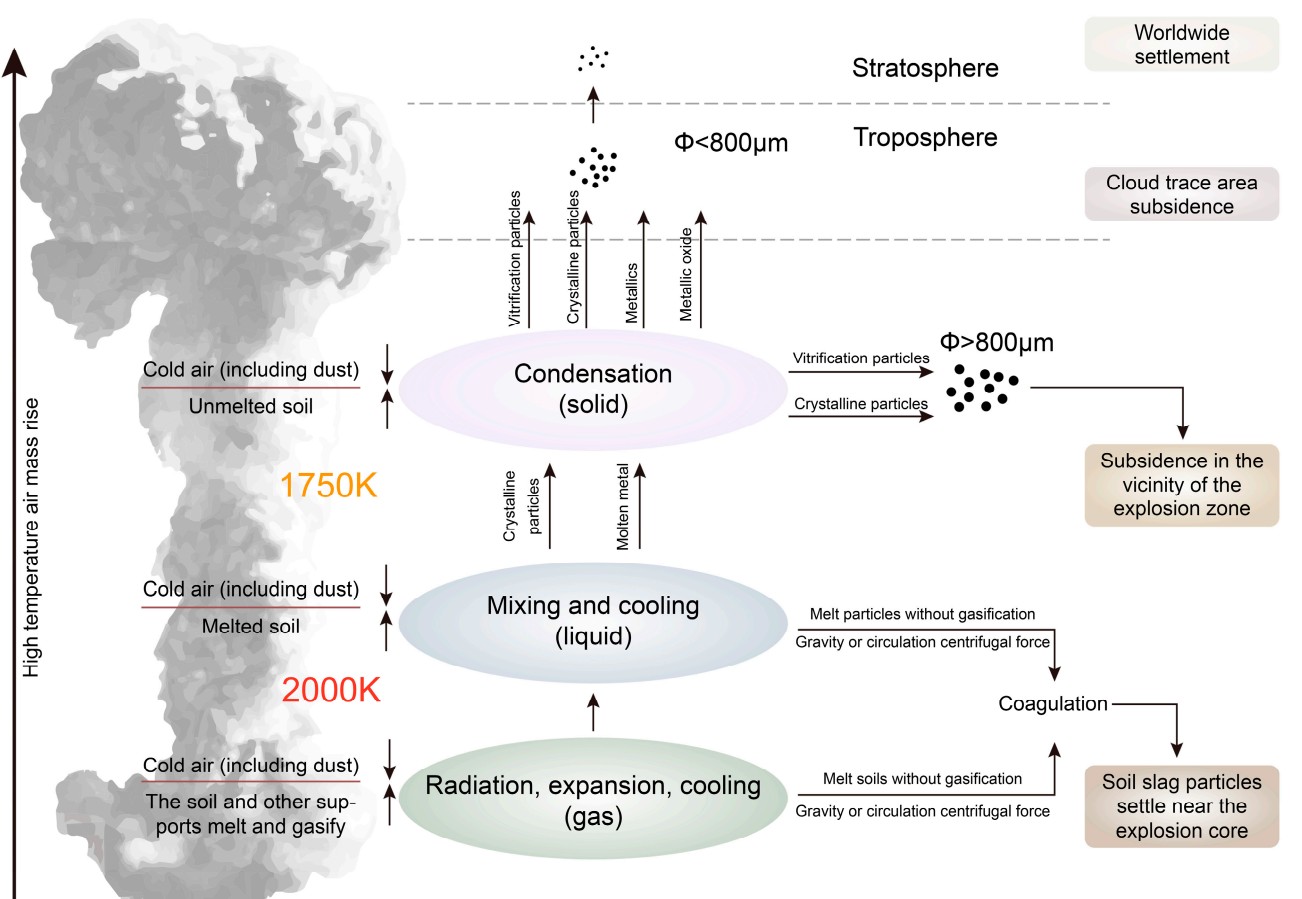

**Figure 3.** The formation process of nuclear explosion radioactive smoke cloud stratification and particles.

*2.4. Implementation Process of GA in Nuclear Weapon Explosion Source Term Parameter Identification*

GA (genetic algorithm) was first proposed by Professor J. Holland in 1975. On the basis of natural selection and the genetics mechanism of Darwinian biological evolution,

the algorithm simulates the natural law of survival of the fittest and evolves the optimal solution of the problem by inheritance, mutation, crossover, and replication from generation to generation starting from the initial population [38,39]. GAs have been widely used in the fields of combinatorial optimization, machine learning, signal processing, and artificial intelligence and are key techniques in modern relevant intelligent computing [39,40]. The implementation of GA has a relatively simple idea and has a broader applicability. The data optimization computational flow of GA is shown in Figure 4.

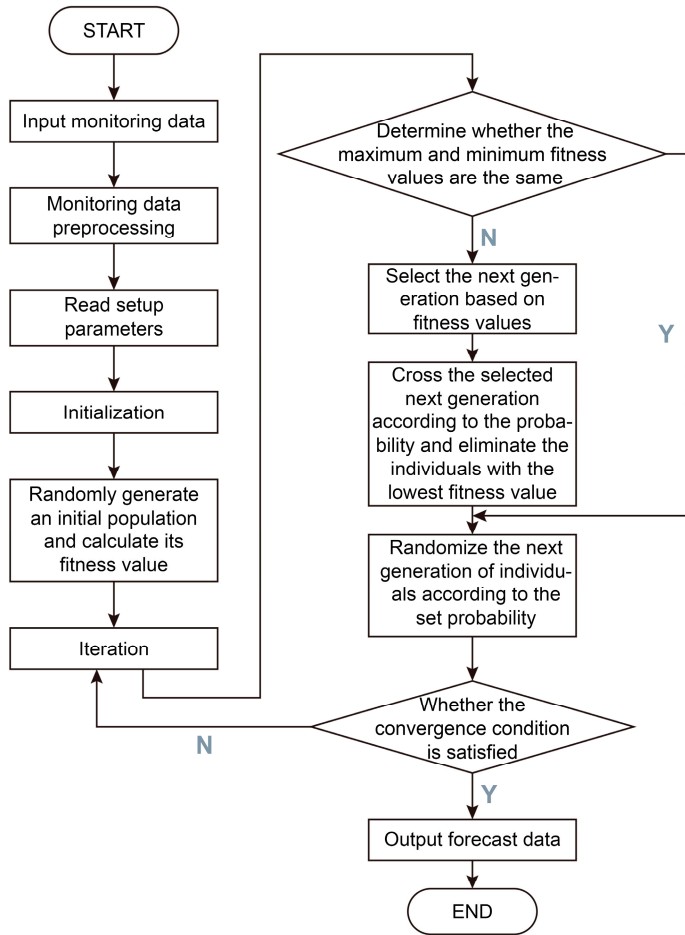

**Figure 4.** Genetic algorithm source term parameter identification calculation flow chart.

### 2.4.1. Construction of Genes and Chromosomes

The radiation level in the ground contaminated area at a certain moment after the explosion of a nuclear weapon can be considered as a function of five key parameter variables: the location of the nuclear explosion center, the explosion yield, the explosion height, the average wind speed and the average wind direction. The feasible solutions constituted by these parameters are referred to as chromosomes in genetic algorithms. These feasible solutions consist of multiple variable elements, and each variable element is then referred to as a gene that makes up the chromosome. The range of values of the source parameters corresponding to the primary chromosomes is shown in Table 1, and the True-Values plus Gaussian errors are entered as the range of values of the source parameters by using the US nuclear test parameter code-named "KOON" as the "True-Value".

### 2.4.2. Construction of the Fitness Function

In nature, according to the law of natural selection, the better individuals in each generation can be selected and some individuals with poor environmental adaptation can be eliminated. In genetic algorithms, it is necessary to construct an adaptation function to measure the superiority and inferiority of chromosomes. In the problem of nuclear weapon

explosion radiation dose prediction, the chromosome is the input parameter of the nuclear explosion radiation hazard calculation model, and the model calculates the radiation level in the ground contaminated area at a certain moment after the nuclear explosion, and the accuracy of the calculation result is the measure of the merit of the chromosome. Therefore, here the correlation coefficient is calculated by using the monitoring value read in Figure 1 and the predicted value of the model calculation result; the larger the correlation coefficient, the better the adaptation. The correlation coefficient is calculated as follows:

$$Corr = \frac{Cov(Y_{dist}, Y_0)}{\sigma(Y_{dist}) \cdot \sigma(Y_0)} \tag{3}$$

where *Cov* stands for covariance and $\sigma$ stands for standard deviation. A higher correlation coefficient means that the observed data are more consistent with the model prediction results.

The value range of the correlation coefficient is $[-1, 1]$, and the values of the correlation coefficient are mapped to the interval $[0, 100]$ to construct the fitness of the chromosome, where the fitness of the chromosome *Fitness* is calculated as

$$Fitness = 50 \cdot (Corr + 1) \tag{4}$$

### 2.4.3. Crossover, Mutation and Selection of Genetic Operators

Each iteration of the GA generates N chromosomes, and each iteration is called an "evolution". The new chromosomes generated at each evolution are generated through a "crossover operation". The crossover operation usually interchanges some chromosomes from a sample of both parents selected from the current generation to create two new chromosomes representing the offspring. This operation is called crossover or recombination, and the exact value of the optimal crossover probability $\omega$ shall be found experimentally. In this article, $\omega$ was chosen as 0.60, 0.65, 0.70, 0.75, 0.80, 0.85, 0.90.

After each completed evolution, the fitness of each chromosome is calculated, and then the fitness probability of each chromosome is calculated using Equation (5). When the crossover process is performed, it is then necessary to select the parent chromosome based on this probability. The higher the probability that chromosomes with higher fitness will be selected, the genetic algorithm is able to retain the better genes.

$$P_{ci} = \frac{Fitness(i)}{\sum_{i=0}^{M-1} Fitness(i)} \tag{5}$$

where $P_{ci}$ is the probability of chromosome *i* being selected and *Fitness(i)* is the fitness of chromosome *i*.

Crossover can ensure that each evolution leaves good genes, but it only selects the original result set. The genes are still the original randomly generated initial values, which keep exchanging the order of permutation between them. This can only ensure that after multiple evolutions, the computational result is closer to the local optimal solution, but it is difficult to reach the global optimal solution. To solve this problem, mutation operations need to be introduced. The purpose of "mutation" is to periodically update the population randomly, introduce new patterns into the chromosome, and perform searches in unknown regions of the solution space. The specific chosen value of mutation probability should be found experimentally as well. The experimental values selected in this paper are 0.005, 0.01, 0.02, 0.05, 0.1.

The operation to select the superior individuals from the population and eliminate the inferior ones is called selection. The selection operation is based on a fitness assessment of the individuals in the population. The genetic operators will be subjected to optimization search and selection experiments to obtain the optimal operator.

(1) Roulette-wheel selection is a method that calculates the probability of the occurrence of each individual in the offspring based on the fitness value of the individual and randomly selects individuals according to this probability to form the offspring population,

so the method is also known as the fitness proportion method [41]. The starting point of the roulette-wheel selection strategy is that individuals with better fitness values have a higher probability of being selected.

In order to calculate the selection probability $P(i)$, the fitness value $f_i$ for each individual $i$ is used:

$$P(i) = \frac{f_i}{\sum_{i=0}^{M-1} f_j} \tag{6}$$

Selecting M individuals from a given population is equivalent to rotating the roulette wheel M times, and it is not actually necessary to sort the individuals in the population before selecting them. Note that Equation (6) assumes by default that all fitness values are positive and that the sum of fitness values is not 0. To be able to handle negative fitness values, it can be improved to Equation (7) to calculate the selection probability.

$$P'(i) = \frac{f_i - f_{min}}{\sum_{i=0}^{M-1} (f_j - f_{min})} \tag{7}$$

Let the worst fitness value be $f_{min}$, and if it is negative, then the selection probability is 0.

(2) Tournament selection operator randomly selects $n$ individuals from the population with replacement, then selects the best individual to enter the next generation. Only the individual whose fitness value is better than the other $n-1$ competitors can win the tournament, and the worst individual will be directly excluded [42]. Selection pressure can be changed by varying the size of the tournament, $n$. When this has a larger value, the weaker individuals have less chance of being selected. Compared with roulette selection, tournament selection is more commonly used in practice due to the lack of random noise and its independence from the scale of the genetic algorithm fitness function.

(3) Truncation selection is an operator that ranks the individuals in a population in order from best to worst according to the fitness value, and the best individual is selected to enter the next generation [43]. The advantage of truncation selection is the ability to quickly select individuals from a large number of populations. Truncation selection is a standard method used in plant and animal breeding, where only the best animals are selected for breeding and they are ranked according to their performance values.

*2.5. Implementation Process of PSO in Nuclear Weapon Explosion Source Term Parameter Identification*

PSO was proposed by Eberhart and Kennedy in 1995 [44]. This algorithm is a global optimization evolutionary algorithm, which is a very effective and widely used iterative optimization algorithm [45,46]. Compared with GA, PSO has the advantage of being computationally simple and fast running, without many parameters to be adjusted, and its unique memory function allows it to dynamically track the current search situation to adjust the search strategy [47,48].

The computational flow of the particle swarm optimization data is shown in Figure 5. The core idea of PSO is as follows:

$$V_i^{t+1} = \omega \cdot V_i^t + c_1 \cdot r_1 \left( P_i^t - X_i^t \right) + c_2 \cdot r_2 \left( G^t - X_i^t \right) \tag{8}$$

$$X_i^{t+1} = V_i^{t+1} + X_i^t \tag{9}$$

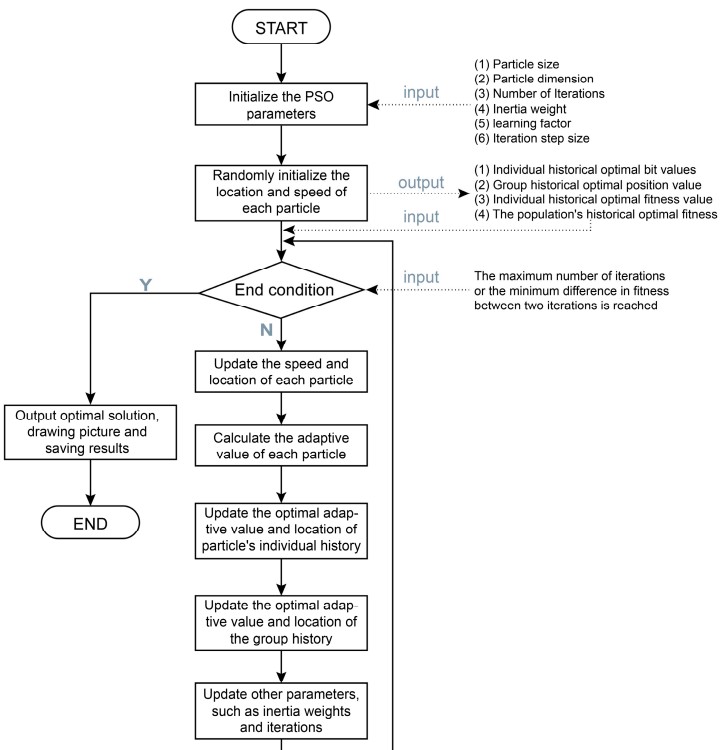

**Figure 5.** Particle swarm optimization algorithm source term parameter identification calculation flow chart.

$V_i^{t+1}$ denotes the velocity of the particle $i$ in $t+1$ iterations, $\omega$ is the inertia weight, which controls the effect of its own velocity on the velocity at the next moment and is used to balance the exploration and exploitation ability of the algorithm. $c_1$ is the learning factor of the particle advancing to the individual historical optimum $P_i$, and $c_2$ is the learning factor of the particle advancing to the global historical optimum $G$. $r_1$, $r_2$ are random numbers between 0 and 1. The nuclear weapon explosion source term parameters are mapped into each random particle, and the particles search for new solutions by continuously adjusting their positions.

2.5.1. Parameter Selection and Optimization of PSO

(1) Particle population size: N

The particle population size N is the number of particles in the population involved in the calculation, and in this experiment is the number of solutions involved in the optimization. In this experiment, the particle population size N is taken to be 100.

(2) Particle dimension: D

Particle dimension D is the spatial dimension number of the particle search is also the number of independent variables, nuclear weapons explosion to be optimized source parameters contain 5 variables: the explosion equivalent *W*, explosion *x*-coordinates, explosion *y*-coordinates, equivalent wind speed *v* and average wind direction *φ*, so experiment D is set to 5.

(3) The number of iterations: K

The number of iterations generally needs to be adjusted according to the actual situation in the process of optimization, too small a K value will lead to non-convergence, instability or fall into the local optimal solution; and too large a K value will consume arithmetic power and take too much time. In this experiment, a typical number of iterations was chosen, and K was taken as 100.

(4) Inertia weight: $\omega$

The inertia weight $\omega$ indicates the influence of the speed of the previous generation of particles on the speed of the contemporary particles. The particles move inertially based on

their own speed, and the inertia weight is the tendency to make the particles maintain the inertia of motion and search the extended space. When the problem space is large, in order to achieve a balance between search speed and search accuracy, it is a common practice to have a high global search capability in the algorithm in the early stage to get a suitable seed and a high local search capability in the later stage to improve the convergence accuracy, so $\omega$ should not be a fixed constant.

In this experiment, the method of taking inertia weights is improved by adopting an adaptive adjustment strategy, i.e., decreasing the value of $\omega$ linearly as iterations proceed, so that the particle swarm algorithm has a strong global convergence ability in the early stage and a strong local convergence ability in the later stage.

$$\omega = \omega_{max} - \frac{i(\omega_{max} - \omega_{min})}{K} \tag{10}$$

where $\omega_{max}$ represents the maximum inertia weight, $\omega_{min}$ represents the minimum inertia weight, $i$ is the current number of iterations, and $K$ is the maximum number of iterations. In this experiment, the initial inertia weight $\omega$ is used as the parameter to be optimized and the experimental values are 0.4, 0.6, 0.8, 1.0, 1.2, 1.4, 1.6.

(5) Learning factors: $c_1, c_2$

The learning factor is also called the acceleration factor. $c_1$ indicates the weight of the particle's next action from its own experience and the acceleration weight that pushes the particle to the individual optimal position Pi; $c_2$ indicates the weight of the particle's next action from the experience of other particles, and the acceleration weight that pushes the particle to the global optimal position G. The specific values of the learning factors take different values for different problems, and the two values are generally adjusted by "trial and error" in an interval. Typical values of the learning factors in this paper are as follows:

$$\begin{cases} C_1 = 2.0, C_2 = 2.0 \\ C_1 = 1.6, C_2 = 1.8 \\ C_1 = 1.6, C_2 = 2.0 \\ C_1 = 2.0, C_2 = 1.8 \end{cases}$$

2.5.2. Initializing the Particle Swarm and Running the PSO Algorithm

The particle swarm is initialized according to the spatial dimension of the independent variable D. The initial range of the parameters of each dimension representative is the same as the range of the initial chromosome in the genetic algorithm. Use the US nuclear test parameter code-named "KOON" as the "True-Value", add the True-Value to the Gaussian error, and use the sum got as the input range of the source term parameters of the initial particle representatives, as shown in Table 1.

The calculation method of particle fitness in the particle swarm algorithm is similar to that of the genetic algorithm: the objective function of data assimilation is constructed according to the closeness of the observed data to the model calculation results, and their correlation coefficients are selected as the objective function for iterative optimization. The correlation coefficient *Corr* is calculated by Equation (11), and the fitness of the particle *Fitness* is obtained from Equation (4).

$$Corr = \frac{Cov(Y_{dist}, Y_0)}{\sigma(Y_{dist}) \cdot \sigma(Y_0)} \tag{11}$$

where *Cov* represents the covariance and $\sigma$ represents the standard deviation.

The initial particle fitness values calculated by the above Equation. Then the PSO algorithm is run for K iterations by updating the velocity and position of each particle, and the final radiation dose rate distribution and the corresponding source term parameters are obtained by optimizing the objective function using the particle swarm algorithm.

*2.6. Evaluation Method*

In order to evaluate the results of the model optimization, the results of the source term parameters represented by the chromosomes generated by each generation of the genetic algorithm or the random particles generated by each generation of the particle swarm algorithm are substituted into the nuclear explosion radioactive contamination prediction model to obtain the corresponding predicted dose values, and then compared with the monitoring values shown in Figure 1, as shown in Figure 6a,b. The model predicted values are then plotted against the monitoring values, as in Figure 6c. Finally, mean absolute error (MAE) $\mu$ and mean squared error (MSE) $\sigma$ are calculated as the prediction evaluation indexes using the following formulae:

$$\mu = \frac{1}{N} \sum_{i=1}^{N} |\mu_i - \hat{\mu}_i| \tag{12}$$

$$\sigma = \frac{1}{N} \sum_{i=1}^{N} (\mu_i - \hat{\mu}_i)^2 \tag{13}$$

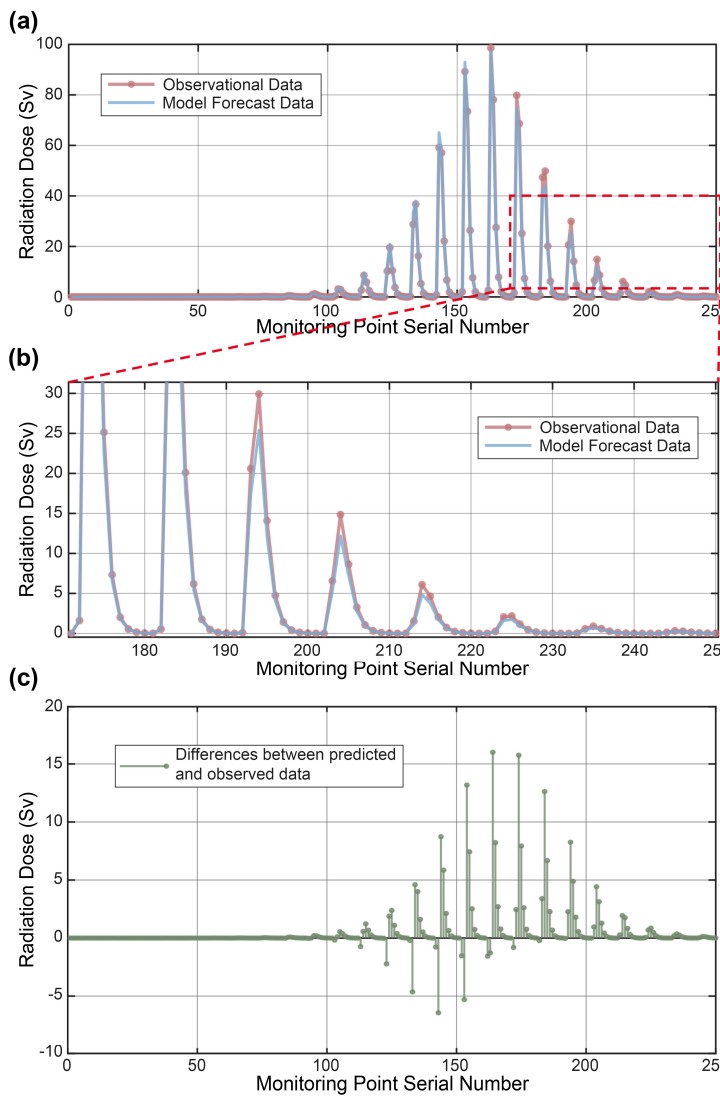

**Figure 6.** Evaluation indicators of the calculation results of the optimization identification algorithm. (**a**) comparison graph of optimized identification calculated dose values with real measurement data for each monitoring site; (**b**) local magnification effect (dashed part of the above figure); (**c**) difference between optimized s identification calculation results and real measurement data.

## 3. Results

### 3.1. Results of GA in Source Term Parameter Identification

#### 3.1.1. Selection of Genetic Operators

As can be seen by the selection experiments with different cross-genetic operators in the genetic algorithm under the same conditions, the initial 100 chromosome fitness for roulette, tournament, and truncated selection all tend to be the same (Figure 7a–c), with the horizontal coordinates of the plots indicating each chromosome, and the vertical coordinates indicating this initial fitness. This is because most of the individual adaptations of the randomly generated primaries are around 50, except for individual chromosomes. All three genetic operators evolved 100 times, and each evolution generated 100 chromosomes (Figure 7d–f). The horizontal coordinate in the figure indicates the number of evolutions, and the vertical coordinate indicates the fitness of this generation.

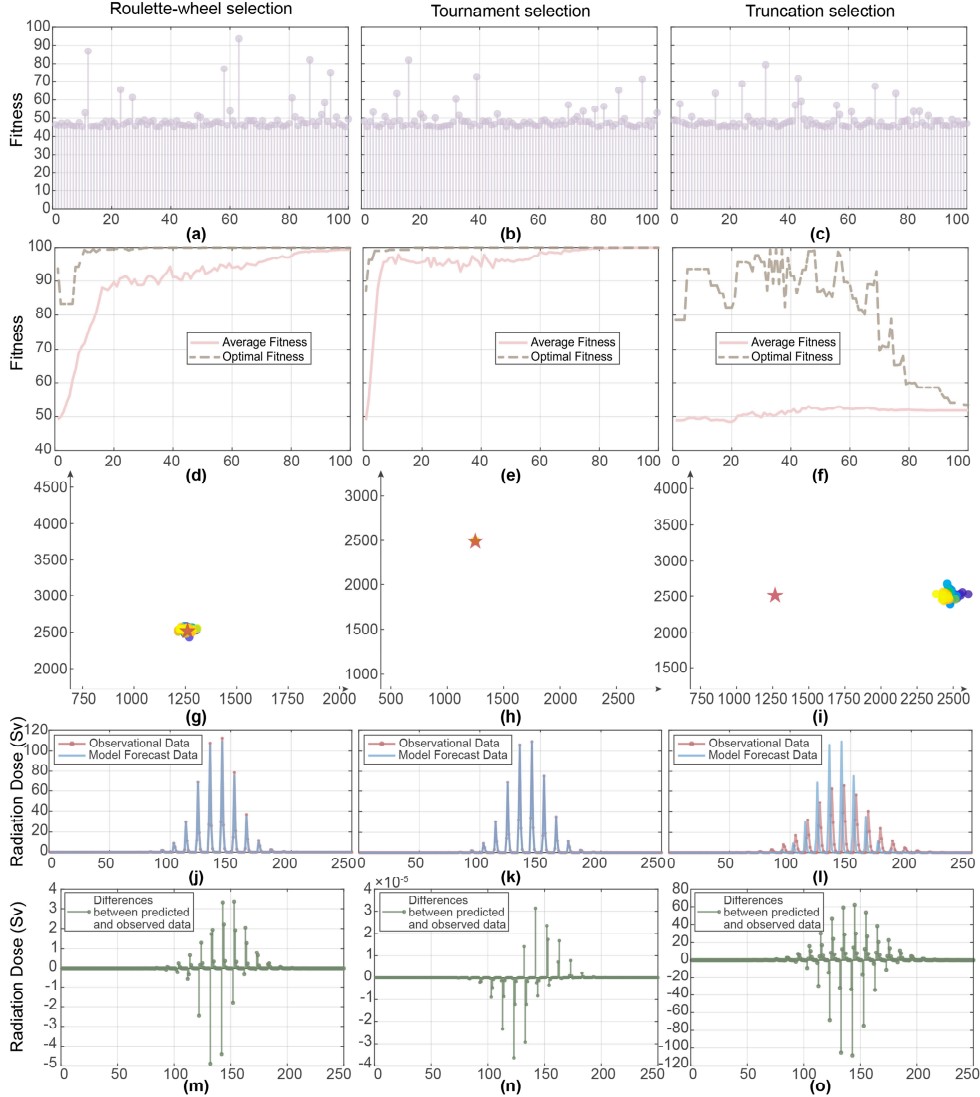

**Figure 7.** Comparison of optimization results of genetic operators. (**a–c**) initial fitness of populations of three genetic operators; (**d–f**) optimal fitness curves of three genetic operators; (**g–i**) 100th generation population evolution results of three genetic operators, the dots represent each random solution and the pentagrams represent the true values; (**j–l**) comparison of calculated dose values for optimal identification of monitoring loci of three genetic operators with real measurement data; (**m–o**) comparison of calculated results for optimal identification of three genetic operators with the difference between the real measurement data.

From the figures, we can see that both the tournament selection operator and the roulette selection operator have good convergence, and the algorithm chromosomes gradually converge to the optimal solution when evolving about 20 times (Figure 7d,e), whereas the fitness of the truncated selection operator does not converge (Figure 7f). After 100 generations of iterations, their particle coordinate trajectory plots are represented as the optimization results of the exploding heart coordinates (Figure 7g–i), and it is obvious that both the roulette operator and tournament operator both converge to the global optimal solution representing the true value, whereas the truncated selection operator falls into the local optimal solution. Further observing the error data of the model dose output values after 100 generations of optimization with respect to the true monitored values (Figure 7j–o), the error of the truncated selection operator is significantly higher than that of the roulette operator and the tournament operator. In contrast, the optimization effect of the tournament operator is higher, so the tournament operator was chosen for this study.

### 3.1.2. Model Parameter Optimization Results

The choice of crossover probability $\omega$ and variance probability are the parameters to be found in the optimization of genetic algorithm in the identification of nuclear weapon explosion source term parameters, and their optimal choice should be concluded by traversing the experiments. Because the results of GA are inherently random, in this paper five repeated experiments were selected to eliminate the random error caused by batch efficiency, and the results are shown in Figure 8.

The MAE and MSE of the experimental results after 100 iterations of each experiment are taken as negative logarithms, and the matrix bubble plots (Figure 8a,b) and grouped box line plots (Figure 8c,d) are plotted, respectively. It is not difficult to find that the choice of the crossover probability $\omega$ and variance probability parameters in the genetic algorithm has no order of magnitude advantage on its optimization accuracy, but the results are very different under the same parameter selection experiment (Figure 8), which again proves that the randomness effect of the genetic algorithm results itself has a greater impact on the final results. When the crossover probability $\omega = 0.08$ and the variance probability is taken as 0.01, the source term results of its model output optimization are relatively stable and accurate, so the genetic algorithm in the nuclear weapon explosion source term parameter identification its crossover probability $\omega$ and variance probability with seeking parameters are taken as 0.08 and 0.01, respectively.

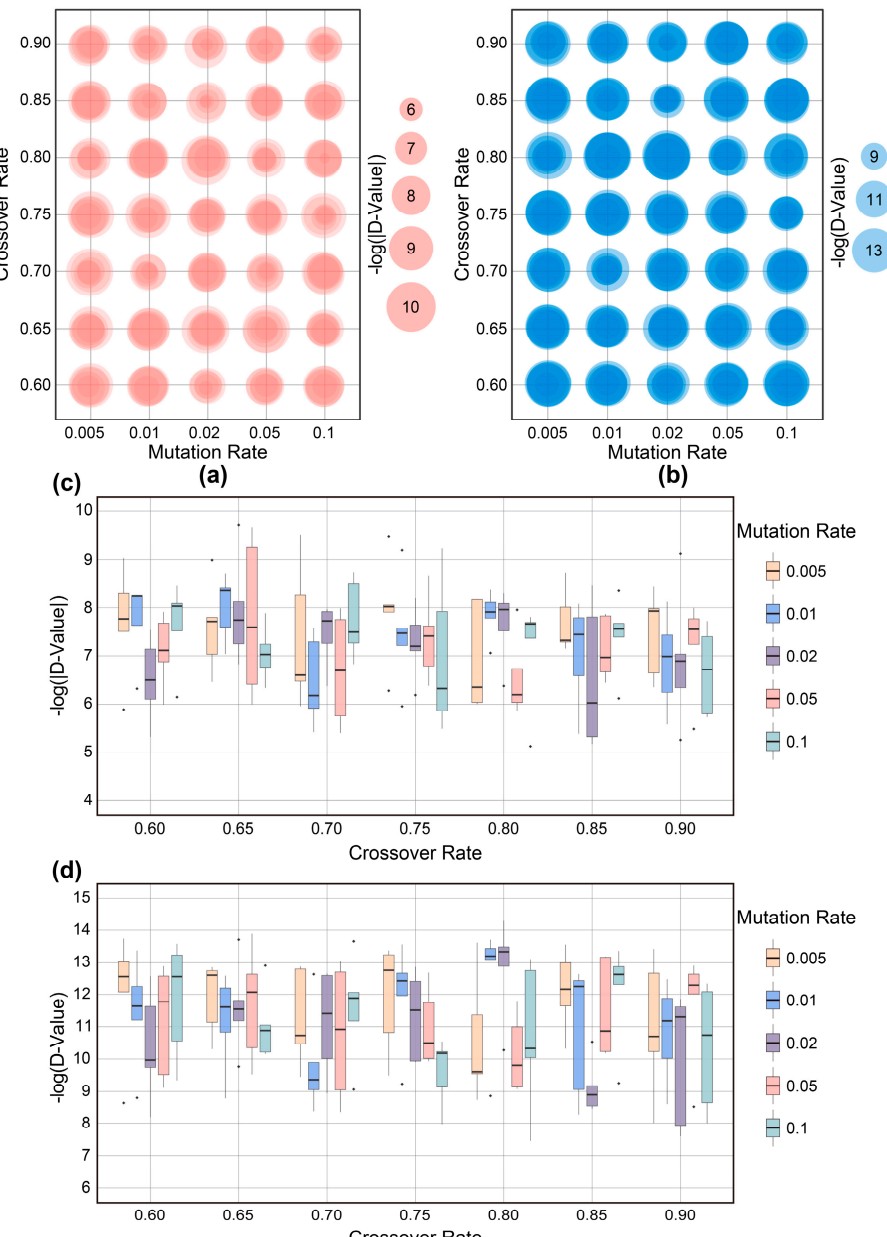

**Figure 8.** Genetic algorithm optimized identification parameter finding results (five replicate experiments). (**a**) MAE bubble plots of optimized identification data with real measurement data for different parameter values; (**b**) MSE bubble plots of optimized identification data with real measurement data for different parameter values; (**c**) MAE grouping boxplots of optimized identification data with real measurement data for different parameter values; (**d**) MSE grouping boxplots of optimized identification data with real measurement data for different parameter values.

### 3.1.3. Analysis of GA Results

The operation process of the genetic algorithm is shown in Figure 9, which visualizes the explosion center coordinates of the source term parameters to be optimized in this model (Figure 9a–f). The chromosomes generated randomly in the initial generation are shown to be randomly distributed in the two-dimensional plane, but the genetic algorithm retrospectively computed after the above optimal parameter selection has a significant convergence effect after 20 iterations (Figure 9b,g).

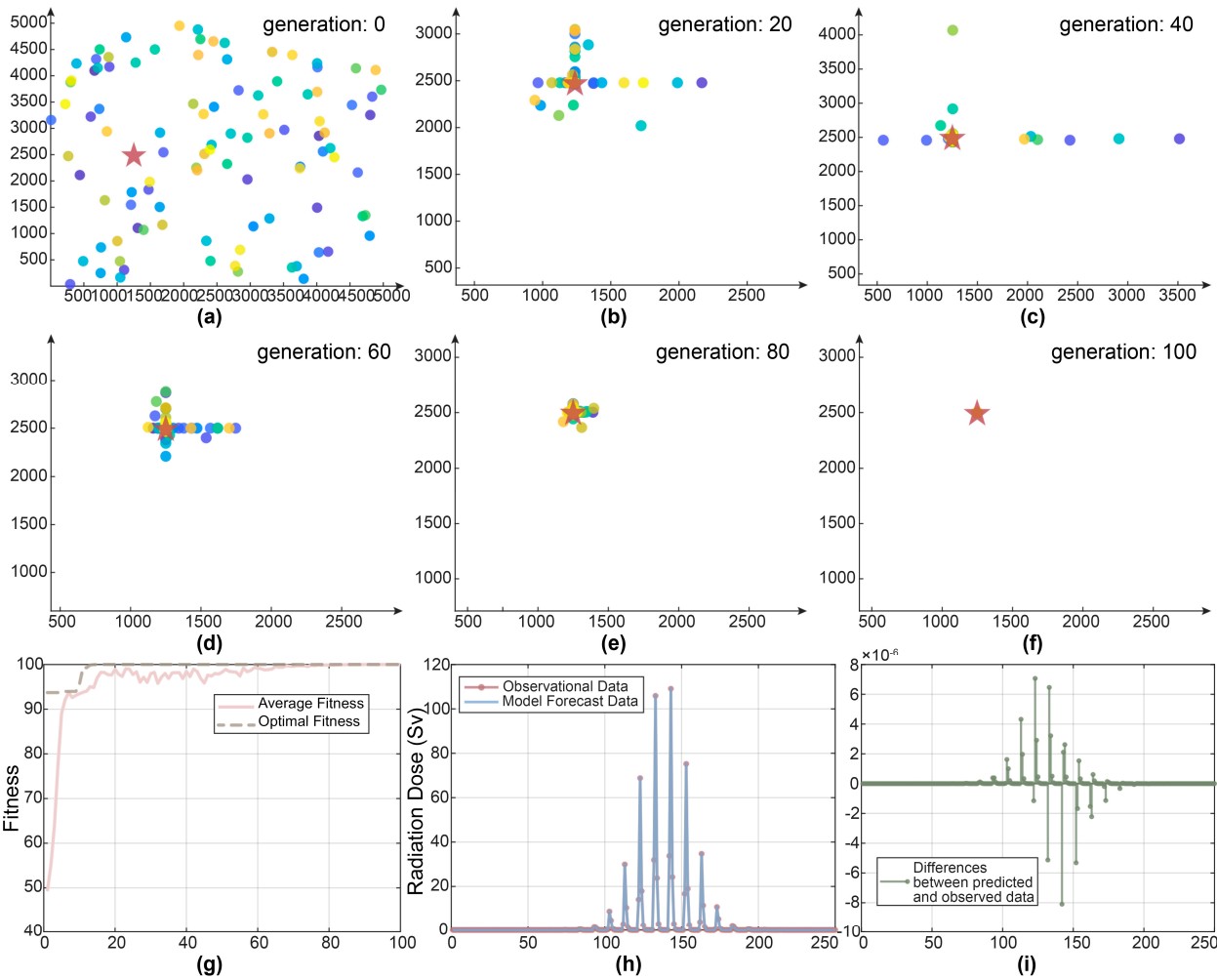

**Figure 9.** Plot of the results of the genetic algorithm optimized identification computational model. (**a–f**) Evolutionary process of the population in the 0th–100th generations, the dots represent each random solution and the pentagrams represent the true values; (**g**) change curve of the average fitness of the population in each generation; (**h**) comparison of the calculated dose values of the optimal identification of the monitoring sites with the real measurement data; (**i**) difference between the calculated results of the optimal identification and the real measurement data.

The adaptation value reflects the degree of agreement between the optimal chromosome after assimilation and the simulated measurement data, and after 100 iterations of the genetic algorithm, its adaptation value is very close to 100 (Figure 9f–h), which is consistent with the fact that this simulated measurement data is calculated from the nuclear explosion radiation hazard model, and secondly, with the adaptation value very close to 100, the ability of the genetic algorithm to search for the optimal value in the whole domain is very good, which shows that it is feasible to use this algorithm for the identification of nuclear explosion source term parameters. Figure 9i shows the results of the forward calculation based on the optimized source term parameters after the traceability using this method, and the agreement with the monitored values is very good, with the statistical mean absolute error $\mu = 5.1674 \times 10^{-8}$ and mean squared error $\sigma = 1.1425 \times 10^{-12}$ between them.

### 3.2. Results of PSO in Source Term Parameter Identification
#### 3.2.1. Model Parameter Optimization Results

The selection of the initial inertia weight $\omega$ and the learning factors $c_1, c_2$ are the parameters to be searched for by the particle swarm algorithm in the optimization of the nuclear weapon explosion source term identification, and their optimal selection should

be concluded by traversal experiment. Similar to the genetic algorithm treatment, five iterations of the experiment were used to eliminate the random error caused by the batch benefit, and the results are shown in Figure 10. The MAE and MSE of the experimental results after 100 iterations of each iteration were taken as negative logarithms and plotted in matrix bubble plots (Figure 10a,b) and grouped box line plots (Figure 10c,d), respectively.

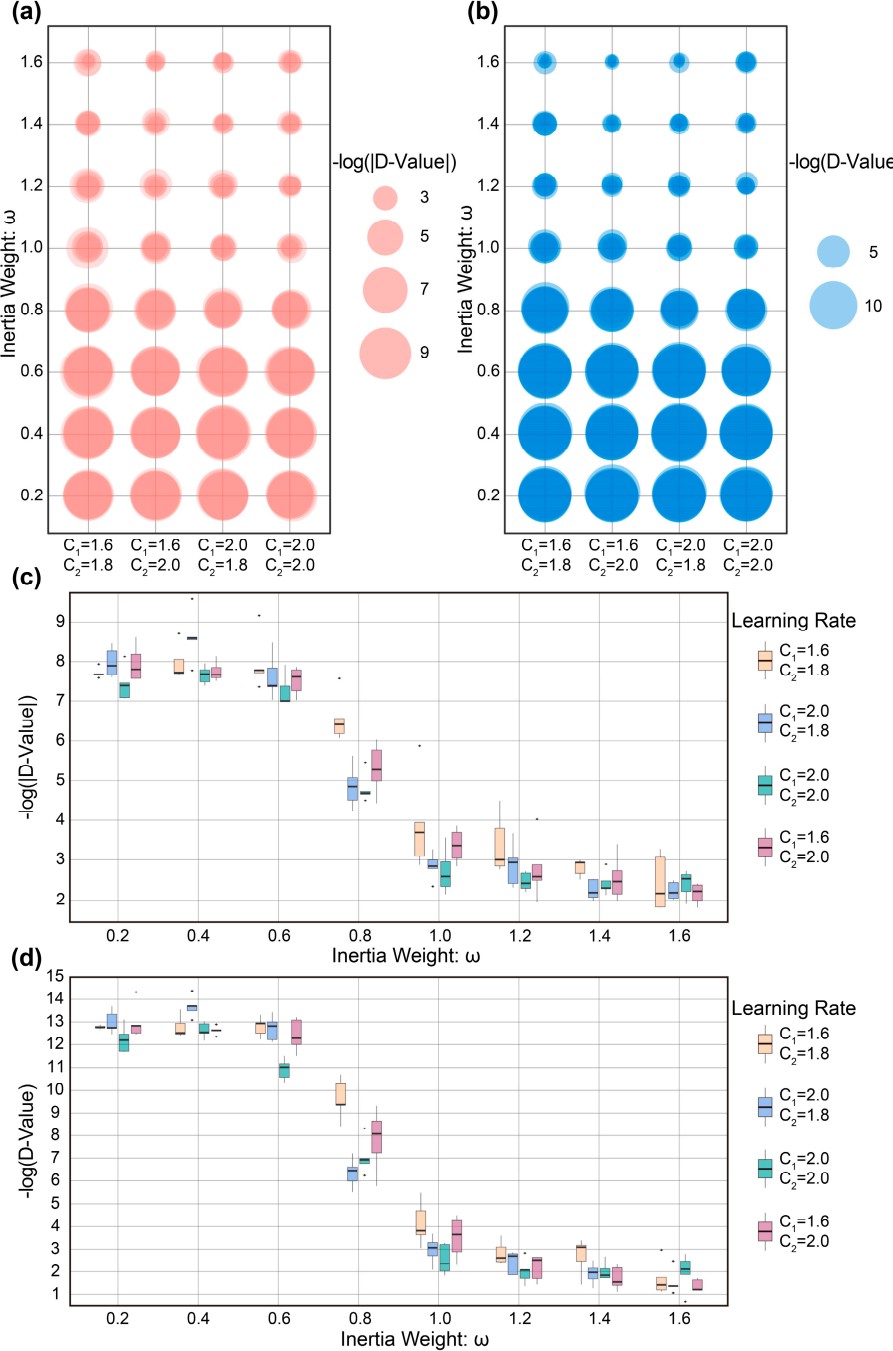

**Figure 10.** Particle swarm optimization algorithm optimized identification parameter finding results (five replicate experiments). (**a**) MAE bubble plots of optimized identification data with real measurement data for different parameter values; (**b**) MSE bubble plots of optimized identification data with real measurement data for different parameter values; (**c**) MAE grouping boxplots of optimized identification data with real measurement data for different parameter values; (**d**) MSE grouping boxplots of optimized identification data with real measurement data for different parameter values.

Compared with the parameter search results of the genetic algorithm, the parameter search results of the particle swarm algorithm are more intuitive, and the overall results are less random. The choice of learning factors has little effect on the optimization accuracy, but from the negative logarithm of the mean absolute error (MAE) and variance (Figure 10), the results are significantly better than 1.0–1.6 when the initial inertia weight $\omega$ is taken as 0.2–0.6. In summary, the initial inertia weight $\omega = 0.4$ was chosen, and the learning factors were chosen as $c_1 = 2.0, c_2 = 1.8$ after the parameter optimization of the particle swarm algorithm.

### 3.2.2. Analysis of PSO Results

After the particle swarm parameter optimization, the nuclear weapon explosion source term parameter identification model was re-run with the optimal parameters, and its iterative process is shown in Figure 11. Figure 11a shows similar results to Figure 7a–c, and the initial particle fitness of the particle swarm algorithm is also concentrated at around 50. Figure 11b shows that the optimal position of the swarm keeps approaching the simulated real position in each iteration. Figure 11c shows the change curve of the fitness of the swarm of particles, and the algorithm finally converges to a point close to the true value, where the fitness of the particles is close to 100, i.e., the correlation coefficient between the model calculation results and the observed data is approximately equal to one.

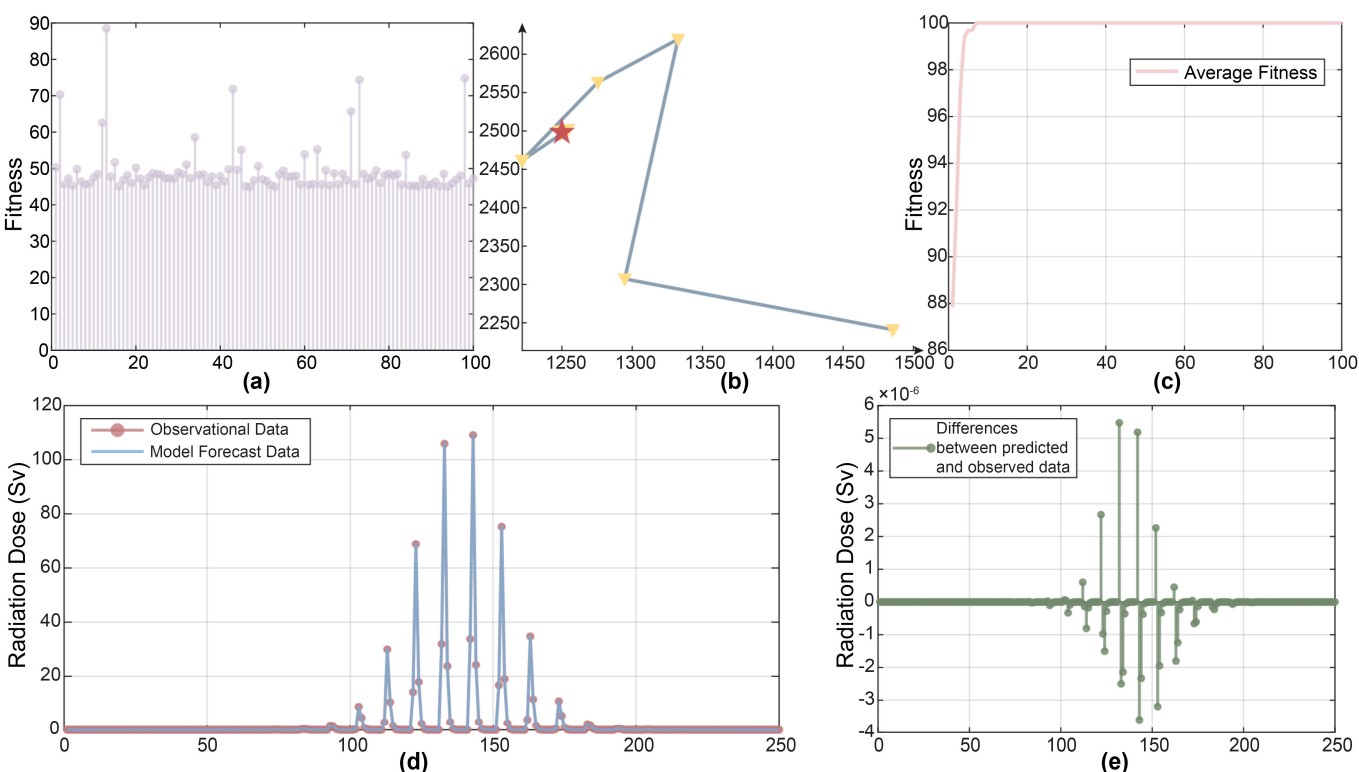

**Figure 11.** Computational model results of optimal particle swarm algorithm identification graph. (**a**) the initial fitness of the optimal particle swarm algorithm; (**b**) the optimal solution search path of the particle swarm algorithm; (**c**) the change curve of the fitness of particles in each generation; (**d**) the results of the optimal source term parameters after optimal identification; (**e**) the difference between the computational results of y optimal identification and the True-Value.

Figure 11d,e are the results of the forward calculation based on the optimized source term parameters after the source term data identification optimization using the particle swarm algorithm. The agreement with the observed data is very good, and the statistical error between them has a mean value $\mu = 4.042 \times 10^{-8}$ and a mean variance $\sigma = 4.9332 \times 10^{-13}$.

### 3.3. Analysis and Comparison of Optimization Results for Source Term Identification

According to the above experimental results, both the genetic algorithm and the particle swarm algorithm can identify and optimize the source parameters of a nuclear weapon explosion using measured data after the corresponding algorithm optimization and parameter search combined with the Lagrange-Gaussian puff model. The fitness for both algorithms converge to 1 after 100 generations of iterative optimization operations, and the results of the forward calculation of the source term parameters identified by both algorithms after optimization are at a high level of agreement with the observed data (Figures 9, 11, and 12).

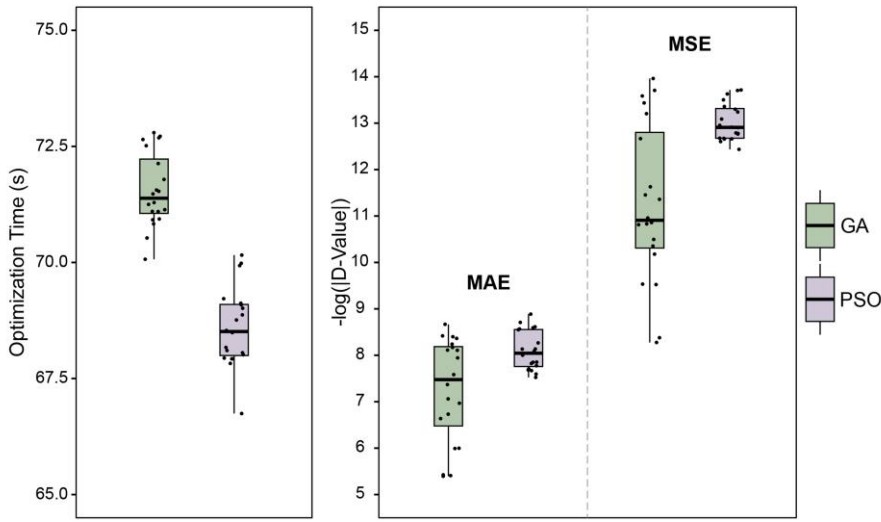

**Figure 12.** Comparison of GA and PSO time consumption and accuracy.

The genetic algorithm and the particle swarm algorithm were applied to this optimized traceability model, the number of iterations of both algorithms was set to 100, and the results of the algorithms and the computing time were compared. Because the swarm intelligence algorithm itself is random, the experiment was repeated 20 times under the same conditions to compare the advantages and disadvantages of the two methods in terms of statistical laws. Finally, it was found that the optimization accuracy and stability of the calculation results of the particle swarm algorithm were slightly better than those of the genetic algorithm, and the calculation time was also slightly shorter than that of the genetic algorithm (Figure 12). The main possible reason for this is that the genetic algorithm itself has more parameters, and the randomness of its calculation process is greater than that of the particle swarm algorithm, so the stability and accuracy of the output results are slightly lower than those of the particle swarm algorithm. The final algorithm output of the nuclear weapon explosion source term parameters after optimal identification of the results can be seen in Table 2. The identification accuracy of the main source term parameters is greater than 98%, and it can be seen that the identification accuracy is satisfactory.

**Table 2.** Real source term parameters with optimized identification.

|  | $W$ *(kt)* | $x_0$ *(m)* | $y_0$ *(m)* | $v$ *(m/s)* | $\varphi$ *(°)* |
|---|---|---|---|---|---|
| Truth-value | 110 | 1250 | 2500 | 6.2 | 191.3 |
| Optimal-value | 109.5843 | 1249.9978 | 2500.0141 | 6.092 | 191.2738 |
| Accuracy Rate | 99.62% | 99.99% | 99.99% | 98.26% | 99.98% |

## 4. Discussion and Conclusions

In this paper, we propose a solution to the problem of inaccurate identification of source term parameters in the prediction model of nuclear weapon explosion radioactive material dispersion: to use the improved GA and PSO algorithms based on the Lagrange-Gaussian puff model to optimize the identification of source term parameters

using real-time monitoring data. The GA and PSO algorithms were used to identify source term parameters with high accuracy by using parameter-finding experiments. The PSO algorithm was combined with the Lagrange-Gaussian puff model, using real historical nuclear explosion data to identify the source term parameters of the "KOON" nuclear test, and the accuracy of its main parameters' prediction was above 98%. The results show that both the particle swarm algorithm and the genetic algorithm were able to identify the nuclear explosion source term parameters more ideally after optimization and maintain high prediction accuracy and precision. When the maximum number of iterations and population size of the particle swarm algorithm and genetic algorithm were the same, the running time and optimization accuracy of the particle swarm algorithm were better than those of the genetic algorithm. This method can provide fast and effective support for nuclear emergency command and decision making in a short period of time and is of great significance for the study of the optimization of the prediction method of the environmental radioactive particle dispersion caused by nuclear weapon explosions.

## 5. Contributions and Limitations

The explosion of a nuclear weapon releases large amounts of radioactive particles into the environment, which, along with the physical movement of the atmosphere, will spread to the area around the explosion (Figure 3) and cause untold effects on the living environment and health of the public. The problem of accurately predicting the radiological hazards of a nuclear weapon explosion has been an enduring topic in the fields of atmospheric science, environmental science, and nuclear physics. A great deal of effort has been spent in previous work to study the rise of nuclear smoke clouds, particle size dispersion and other problems, and remarkable contributions have been made [49–52]. However, no model can achieve absolutely accurate predictions, and the errors introduced by the source term parameters as the input value of any model is one of the important factors affecting the prediction accuracy of the model. In the field of nuclear accidents, the use of algorithms to optimize the identification of source term parameters has achieved many amazing results, but few have been applied in the field of nuclear weapon explosion traceability research. The main purpose of this paper was to try to combine the mature use of the GA and PSO algorithms in nuclear accident source term parameter identification with the classical nuclear weapon explosion radioactive material deposition Lagrange-Gaussian puff model so that the source parameters of a nuclear weapon explosion could be identified and calibrated based on a swarm intelligence algorithm. The results are innovative and informative.

There are also shortcomings in this study, which will be overcome by the ongoing and upcoming work of our team. The main limitations include:

1. Neglecting model errors. The study does not discuss the model error in the Lagrange-Gaussian puff model fitting the prediction of the radioactive smoke cloud of a nuclear weapon explosion itself, and directly assumes that the measured values are equal to the predicted value of the true source term parameters input into the model. However, this error can interfere with the source term parameter identification process. The next study will try to perform an optimization search experiment using algorithms such as GA and PSO in combination with several well-established models such as DELFIC [53] or WSEG [54] so as to analyze the impact of each prediction model error on the source term parameter identification accuracy.

2. Fewer swarm intelligence algorithms are utilized, and there are various classifications of swarm intelligence algorithms, each of which has its own advantages and disadvantages. The next study can try to use more algorithms other than GA and PSO to try to optimize the identification of nuclear weapon explosion source term parameters.

3. Each algorithm itself can be further optimized. In addition to each algorithm's parameter optimization, the next study will explore a combination of optimization of each algorithm and combine each algorithm with machine learning, neural networks and other methods.

**Author Contributions:** Conceptualization, Y.W. (Yuyang Wang) and X.C.; methodology, Y.W. (Yunhui Wu); software, Y.W. (Yunhui Wu), Y.Z. and L.W.; validation, X.L.; formal analysis, M.Y.; investigation, S.J.; resources, X.P. and H.Y.; data curation, W.L.; writing—original draft preparation, Y.Z.; writing—review and editing, Y.W. (Yuyang Wang); visualization, Y.Z.; supervision, L.H. and P.L.; project administration, Y.W. (Yunhui Wu) and X.P. All authors have read and agreed to the published version of the manuscript.

**Funding:** This research received no external funding.

**Institutional Review Board Statement:** Not applicable.

**Informed Consent Statement:** Not applicable.

**Data Availability Statement:** Data used in this study are detailed in Section 2.1. All data used are reflected in the article. If you require other relevant data, please contact the authors.

**Conflicts of Interest:** The authors declare no conflict of interest.

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
