# Peer review of "Identification Method of Source Term Parameters of Nuclear Explosion Based on GA and PSO for Lagrange-Gaussian Puff Model"

_atmosphere, doi:10.3390/atmos14050877_

Round 1

Reviewer 1 Report

MS: atmosphere-2360007

Title: Research on Identification Method of Source Term Parameters of Nuclear Explosion Based on GA and PSO for Lagrange-Gaussian Puff Model has been proposed by the authors. This study enriches the theory and method of radioactive particle dispersion prediction after nuclear weapon explosion, and is of great significance to the study of environmental radioactive particles. This article can be included in “atmosphere” after minor revisions.

1. This study is incorrect in writing format. Please check the format at “atmosphere”.

2. The format of brackets, en dash, hyphen, comma (), and tilde (~) did not follow the common usage of journal.

3. The styling of Tables and writing using track change is messy.

4. The resolution of most Figures are too low to print for publisher.

5. Some Figures should include the reference.

6. To make Tables more understandable, I advise the authors to rebuild them.

7. On section 3.1, the description of methods and study area is relatively lacking. It is recommended to add such as features or other references.

8. It is recommended to confirm the results of the simulation to ensure the accuracy of the results.

9. A few grammatical and typo mistakes are still there in your manuscript. Please thoroughly check and revise the manuscript carefully.

10. All units should be adopted according to consistent units.

11. Literature writing format of this article is confused.

12. The content is plentiful, but some part of the reference literatures is kind of obsolete (in 5 years). Key publications should be cited as completed as possible. Please also clarify the novelty and application implication of your work in this section. I suggest authors refer to the latest literatures from “atmosphere”, “MDPI”, and other disaster risk journals. But please do not exceed 30% of all citations from atmosphere. Authors may see the following reference while revising. https://doi.org/10.1016/j.jlp.2023.105030;  https://doi.org/10.1016/j.jlp.2023.104977; https://doi.org/10.1007/s10973-023-12012-8; https://doi.org/10.1007/s10973-022-11898-0

Author Response

-------------------------------------

Response to Reviewer 1

-------------------------------------

General comment:

Research on Identification Method of Source Term Parameters of Nuclear Explosion Based on GA and PSO for Lagrange-Gaussian Puff Model has been proposed by the authors. This study enriches the theory and method of radioactive particle dispersion prediction after nuclear weapon explosion, and is of great significance to the study of environmental radioactive particles. This article can be included in “atmosphere” after minor revisions.

Response: We thank the reviewer for the positive evaluation for the manuscript and for pointing out our problems. All the suggestions and questions are valuable and help improve the overall quality of this work. We sincerely and humbly accept these kind reminders.

Major comments:

  1. This study is incorrect in writing format. Please check the format at “atmosphere”.

Response: Thanks to the reviewers for their detailed suggestions. We have once again conducted a comprehensive proofreading and revision of our manuscript in accordance with the writing standards and requirements of "atmosphere".

  1. The format of brackets, en dash, hyphen, comma (), and tilde (~) did not follow the common usage of journal.

Response: Thanks to the reviewer for this suggestion. After careful examination, we have standardized the entire text on the usage of the format of brackets, en dash, hyphen, comma (、), and tilde (~).

  1. The styling of Tables and writing using track change is messy.

Response: All tables have been carefully proofread based on the template provided by "atmosphere".

  1. The resolution of most Figures are too low to print for publisher.

Response: The reviewer's suggestion on Figures resolution is very meaningful. We replaced all the figures with higher resolutions to meet the printing needs of the publisher.

  1. Some Figures should include the reference.

Response: In response to this suggestion, we first apologize for our negligence. Figure 1 does indeed come from references, so we have added references to it.

  1. To make Tables more understandable, I advise the authors to rebuild them.

Response: The professional advice of the reviewer is greatly appreciated, we have rebuilt the form of tables, in order to read it easier.

  1. On section 3.1, the description of methods and study area is relatively lacking. It is recommended to add such as features or other references.

Response: The professional advice of the reviewer is greatly appreciated, by referring to the writing methods of other published literature on "Atmosphere" and the writing templates provided by itself, the research methods and results are generally described separately. Therefore, the research methods lacking in section 3.1 are discussed in detail in section 2.4.

  1. It is recommended to confirm the results of the simulation to ensure the accuracy of the results.

Response: Thanks again to the reviewers for their constructive suggestions. The comparison of the optimized results of our algorithm with the real results is shown in table2.

  1. A few grammatical and typo mistakes are still there in your manuscript. Please thoroughly check and revise the manuscript carefully.

Response: Thanks to the reviewers for their suggestions on details. We have once again thoroughly reviewed the grammar and spelling issues in our manuscript.

  1. All units should be adopted according to consistent units.

Response: Thanks for this advice, and we have adopted according to consistent all units and added lacking units from our previous manuscript in Figure 1.

  1. Literature writing format of this article is confused.

Response: Thanks to the reviewers for your comments and suggestions on our writing format. We humbly accept your kind reminder. And with reference to the published articles on 'atmosphere', we have reorganized our literature writing format.

  1. The content is plentiful, but some part of the reference literatures is kind of obsolete (in 5 years). Key publications should be cited as completed as possible. Please also clarify the novelty and application implication of your work in this section. I suggest authors refer to the latest literatures from “atmosphere”, “MDPI”, and other disaster risk journals. But please do not exceed 30% of all citations from atmosphere. Authors may see the following reference while revising.

https://doi.org/10.1016/j.jlp.2023.105030;  https://doi.org/10.1016/j.jlp.2023.104977; https://doi.org/10.1007/s10973-023-12012-8; https://doi.org/10.1007/s10973-022-11898-0

Response: The reviewers were very meticulous in their suggestions for the references in our manuscript. It is worth mentioning that due to the particularity of nuclear weapon explosion experiments. In addition, all countries around the world are currently forbidden conducting atmospheric core tests. Which cause the relevant data very scarce. So, our reference literature cited some relatively ancient historical research results. And the four references recommended by the reviewers are very informative, as they have all been added to my reference list.

I would thank you for your consideration of our manuscript for publication in atmosphere.

Sincerely yours,

Yunhui Wu, Ph.D.

State Key Laboratory of NBC Protection for Civilian, China

Reviewer 2 Report

The manuscript looks fine, and the presentation looks nice to me, but there are several points could be improved.

In figure 1, can the authors explain what those contours mean? There are values of '500, 250, 100...', but these values are not explained. What is the dimension?

Some paragraphs have grammar issues.

1. Line 200~202:

Crossover can ensure that each evolution leaves good genes, but it only selects the original result set, the genes are still the original randomly generated initial values, but just keep exchanging the order of permutation between them.

I suggest to cut this long sentence into a few short sentences with sound grammar logic.

2. Line 211~213:

The selection operation is based on the fitness assessment of the individuals in the population, and this paper to performs the optimal search and selection experiment for the optimal genetic operator.

There are grammar issues within this sentence.

3. Line 528:

but in the field of nuclear weapon explosion traceability research but few

The grammar can be improved.

Overall, I would suggest the authors to read the manuscript thoroughly, and proofread it carefully.

Author Response

-------------------------------------

Response to Reviewer 2

-------------------------------------

General comment:

The manuscript looks fine, and the presentation looks nice to me, but there are several points could be improved.

Response: We thank the reviewer for the positive evaluation for the manuscript and for pointing out our problems. All the suggestions and questions are valuable and help improve the overall quality of this work. We sincerely and humbly accept these kind reminders.

Major comments:

  1. In figure 1, can the authors explain what those contours mean? There are values of '500, 250, 100...', but these values are not explained. What is the dimension?

Response: The professional advice of the reviewer is greatly appreciated. We have added the unit to the annotation in the legend, and it is worth mentioning that due to the long history of this data, the old dose rate unit r/h has been used.

  1. Comments on the Quality of English Language. Some paragraphs have grammar issues.

   1. Line 200~202:

Crossover can ensure that each evolution leaves good genes, but it only selects the original result set, the genes are still the original randomly generated initial values, but just keep exchanging the order of permutation between them.

I suggest to cut this long sentence into a few short sentences with sound grammar logic.

  1. Line 211~213:

The selection operation is based on the fitness assessment of the individuals in the population, and this paper to performs the optimal search and selection experiment for the optimal genetic operator.

There are grammar issues within this sentence.

  1. Line 528:

but in the field of nuclear weapon explosion traceability research but few

The grammar can be improved.

Overall, I would suggest the authors to read the manuscript thoroughly, and proofread it carefully.

Response: Many thanks to the reviewer for their grammar modification suggestions. We have adopted all of them and made modifications in the manuscript.

  1. Line 200~202, we have changed to:

Crossover can ensure that each evolution leaves good genes, but it only selects the original result set. The genes are still the original randomly generated initial values, just keep exchanging the order of permutation between them.

  1. Line 211~213, we have changed to:

The selection operation is based on the fitness assessment of the individuals in the population. The genetic operators will be subjected to optimization search and selection experiments to obtain the optimal operator.

  1. Line 528, we have changed to:

In the field of nuclear accidents, the use of algorithms to optimize the identification of source term parameters has achieved many amazing results, but few applicated in the field of nuclear weapon explosion traceability research.

I would thank you for your consideration of our manuscript for publication in atmosphere.

Sincerely yours,

Yunhui Wu, Ph.D.

State Key Laboratory of NBC Protection for Civilian, China

Reviewer 3 Report

/

Author Response

-------------------------------------

Response to Reviewer 3

-------------------------------------

General comment:

Overall comments: Accept after minor revision. The simulation study is based on the data obtained from source term parameters of the "KOON" nuclear test done on April 6 1954, which is done long time ago with limited measurements and instrument for the data report. The study is to improve the accuracy of source term parameters by adapting the artificial intelligence algorithms of GA and PSO. Overall, the results presentation and data analysis are greatly done with comprehensive explanation and discussion. Some of the limitations stated also covered the major issues and well describe for future studies.

Response: We thank the reviewer for the positive evaluation for the manuscript and for pointing out our problems. All the suggestions and questions are valuable and help improve the overall quality of this work. We sincerely and humbly accept these kind reminders.

It is worth mentioning that due to the particularity of nuclear weapon explosion experiments. In addition, all countries around the world are currently forbidden conducting atmospheric core tests. So, the relevant data is very scarce, we only chose the "KOON" nuclear experiment from limited public information.

Major comments:

  1. Title: suggest to remove ‘Research on’ and start the title with ‘ldentification Method…’ it will directly reflect to the main work of the study.

Response: We are very grateful to the reviewer for their suggestions on the title and have removed‘Research on’, starting with ‘ldentification Method…’.

  1. Abstract: Many well-established models exist for predicting the dispersion of radioactive particles that will be generated in the surrounding environment after a nuclear weapon explosion. -suggest stating some examples of the available models that has been studied.

Response: Thank you very much for the reviewer's suggestions on our abstract section. We humbly accept them by stating some examples of the available models that has been studied and modify this sentence to: The However, without exception, almost all models rely on accurate source term parameters, such as DELFIC, DNAF-1, and so on.

  1. Abstract: Unlike nuclear experiments, accurate source term parameters are often not available once a nuclear weapon is used in a real nuclear strike. -suggest furthering the statement by initial figuring the current problems such as 'due to…'

Response: We thank the reviewers for their suggestions on the abstract again. However, as for this one suggestion, we stick with the original approach. Because the it’s descripted that a nuclear weapon explosion is different from a nuclear accident, but also that being struck by a nuclear weapon is different from conducting a nuclear weapon experiment, we have adopted this emphatic sentence in order to emphasize this difference

  1. Introduction: Many nuclear weapons tests throughout history have caused severe radioactive contamination of indigenous populations near the test sites [12,13], and the surge in the incidence of thyroid cancer and other cancers has further increased the importance of research on radioactive contamination from nuclear explosions -suggest to write as incidence of cancers such as thyroid cancer, xxx……please add more possible cancers example that have been reported from the cited articles

Response: The professional advice of the reviewer is greatly appreciated. We have changed this sentence to “Many nuclear weapons tests throughout history have caused severe radioactive contamination of indigenous populations near the test sites [12,13]. And the surge in the incidence of cancers such as leukemia, lung cancer, thyroid cancer and so on, which has further increased the importance of research on radioactive contamination from nuclear explosions.”.

  1. How reliable and degree of errors for the previous and current work? I think authors need to improve and strengthen this part by a few examples of the GA and PSO applications in simulating some nuclear incidents/explosion if available.

Response: The reviewer's suggestion is indeed highly professional, forward-looking, and constructive. We should really apply our GA and PSO approach to more fully validate it with more experimental data from nuclear accidents or explosions. However, in fact, due to the special nature of nuclear explosion research, there is very limited real data that can be used directly in academic research, so the results of the "KOON" experiment are the most convincing among our existing validations. This is indeed the limitation of our paper, so we also point out that more nuclear weapon test data can be used for verification in future work.

  1. I think authors need to state as limitation of the study and justify it as publicly available data for the nuclear test since it was done long time ago?

Response: Firstly, we are very grateful and humbly accept the reviewer's suggestion. There are indeed some shortcomings in the work of this article, as discussed in section 5. Contributions and Limitations, in which we have already described it. And made corresponding prospects for the future based on our shortcomings. And as mentioned above, due to the particularity of nuclear weapon explosion experiments. In addition, all countries around the world are currently forbidden conducting atmospheric core tests. So, the relevant data is very scarce, we only chose the "KOON" nuclear experiment from limited public information.

I would thank you for your consideration of our manuscript for publication in atmosphere.

Sincerely yours,

Yunhui Wu, Ph.D.

State Key Laboratory of NBC Protection for Civilian, China

Author Response

-------------------------------------

Response to Reviewer 4

-------------------------------------

General comment:

The present manuscript aims to identify the method of source term parameters of nuclear explosion based on GA and PSO for Lagrange-using Gaussian Puff Mode. The research is relevant in the field of well-established models which exist for predicting the dispersion of radioactive particles that will be generated in the surrounding environment after a nuclear weapon explosion. The obtained results show that both the PSO and the GA are able to identify the source term parameters satisfactorily after optimization.

Response: We thank the reviewer for the positive evaluation for the manuscript and for pointing out our problems. All the suggestions and questions are valuable and help improve the overall quality of this work. We sincerely and humbly accept these kind reminders.

Major Comments:

  1. Abstract looks general. Please highlight the salient findings. Also, add the statistical results in the abstract.

Response: Thanks to the reviewers for their comments on the abstract. We have rewritten abstract section by highlighted salient findings and added the statistical results. “The results show that both the PSO and the GA are able to identify the source term parameters satisfactorily after optimization, and the prediction accuracy of their main source term parameters is above 98%.” 98% of this iconic result is significantly reflected here.

  1. Keywords: Avoid the words used in the title.

Response: Thanks for the advice, and we have made adjustment to the keywords to avoid excessive overlap with the title.

  1. Highlight the importance of the aim of the current study. In the introduction part some results from literature must be discussed. Write the novelty of this study before the objectives.

Response: The reviewers' suggestions for the introductory section of our manuscript are of great help and significance, which will make our manuscript completer and more logical. So, we highlight the importance of the aim of the current study by add this sentence “Therefore, it is very necessary to obtain accurate source term parameters through real measurements or optimization algorithms, as well as other means, so as to provide a solid and powerful foundation for the subsequent operations of the prediction model.

And we discussed the results from literature and write the novelty of this study before the objectives by add this sentence “The combination of these optimized algorithms and the measured data can provide the identification accuracy of source term parameters for accidents such as radioactive leaks in nuclear power plants to more than 95%, which provides a reference for research in the field of nuclear weapon explosions.

  1. Mention the number of replications of each test conducted. A number of analyses has been done but, in my opinion, results were not discussed in detail. Depth of analysis is lacking in the paper.

Response: The professional advice of the reviewer is greatly appreciated, and we accept these constructive and helpful suggestions with great humility. In the research experiments covered in this paper, we emphasize the number of repetitions of each experiment, the main purpose of which is to eliminate the random error and the inevitable batch effect of each experiment, which we think is very necessary and very meaningful. One of the purposes of our manuscript, which was written after a brief analysis of the existing results, is to make research ideas and inspiration available to a wider audience and researchers in related fields, thus triggering their deeper research and exploration. At the same time, we would like to get more experts to provide some specific suggestions to improve the study.

  1. Increase the font size in Figures 5-11.

Response: All figures have been redrawn and their resolution has been increased. As suggested by the reviewer, the font size of figures 5-11 has been increased to provide readers with a better reading experience and easier understanding.

  1. The writing of the conclusion section should be improved. Please add the specific findings in the conclusion.

Response: We thank the reviewers for their guidance and suggestions on writing our conclusion section. In our initial manuscript, the conclusion narrative was indeed too brief and unclear. We have now refined it and added more of our experimental results as suggested by the reviewers.

I would thank you for your consideration of our manuscript for publication in atmosphere.

Sincerely yours,

Yunhui Wu, Ph.D.

State Key Laboratory of NBC Protection for Civilian, China

Round 2

Reviewer 4 Report

I agree with publication.